# Antioxidants from *Hyeronima macrocarpa* Berries Loaded on Nanocellulose: Thermal and Antioxidant Stability

**DOI:** 10.3390/molecules27196661

**Published:** 2022-10-07

**Authors:** Andrés Felipe Alzate-Arbelaez, Farid B. Cortés, Benjamín A. Rojano

**Affiliations:** 1Laboratorio Ciencia de los Alimentos, Facultad de Ciencias, Universidad Nacional de Colombia, Sede Medellín, Carrera 65 # 59A-110, Medellín 050034, Colombia; 2Grupo de Fenómenos de Superficie, Michael Polanyi, Departamento de Procesos y Energía, Facultad de Minas, Universidad Nacional de Colombia, Cra. 80 # 65-223, Medellín 050034, Colombia

**Keywords:** anthocyanins, *Hyeronima macrocarpa*, free radicals, berries powder, nanocellulose, storage time

## Abstract

This study investigated the effect of different storage temperatures (35–55 °C) on the bioactive substances and antioxidant properties of *Hyeronima macrocarpa* berries loaded on nanocellulose. NC was extracted from banana pseudo-stems and presented an interesting surface and porosity properties. The acidified ethanol extract showed better anthocyanin extraction (1317 mg C3G eq./100 g FW) and was used for the preparation of the powdered product, which presented an intense and uniform magenta color, with CIELAB parameters of L* = 59.16, a* = 35.61, and b* = 7.08. The powder exhibited significant stability at storage temperatures of 35 and 45 °C, in which there was no significant loss of anthocyanins or a decrease in antioxidant capacity. In addition, the color was stable for up to 4 months without adding any preservative agent. The anthocyanin-rich extract of *H. macrocarpa* reached an estimated shelf-life of 315 days (stored at 35 °C), as a result of the impregnation process between the extract and NC, with the ability to protect the bioactives from degradation, due to NC surface properties.

## 1. Introduction

In South America, various berries are distributed and grow wild. There are species such as the tree grape (*Myrciaria cauliflora*), small blackberry (*Rubus* spp.), *agraz* (*Vaccinium meridionale*), blueberry (*Vaccinium corimbosum*), *arazá* (*Psidium cattleianum*), *jaboticaba* (*Plinia cauliflora*), Cavendishia spp., Disterigma spp., and *coral* blueberry or *motilón* (*Hyeronima macrocarpa*) [1,2,3]. This type of fruit is small, round, and characterized by having an intense red or violet color, with strong flavors, slight acidity, sweet tones, and wide- spread acceptability as a nutraceutical [4].

The main antioxidant bioactive substances in berries are vitamins (A, C, and E) and polyphenolic derivatives such as phenolic acids, tannins, and flavonoids, including anthocyanins, flavones, and flavonols [5]. Polyphenolic compounds are bioactive molecules since they help combat the effects of oxidative stress by preventing or delaying the onset of heart disease, diabetes, and some types of cancer [6,7]. The antioxidant mechanisms in the human body are complex but have been grouped into the following processes: inhibition of lipid peroxidation, elimination of or reduction in free radicals, activation of endogenous antioxidant enzymes, and metal ion chelation [8]. Beyond the multiple benefits of these molecules to maintain an adequate oxidative balance in the body, there are bioavailability and stability considerations, which must be taken into account when ingesting berries or derived products.

Anthocyanins are antioxidant metabolites that, in addition to presenting health benefits, are characterized by their qualities as natural colorants, especially in an acid medium, where they offer colors that cover the scale of oranges, reds, and violets [9]; however, due to their chemical structure, they present a low intestinal absorption that does not exceed 3% [10]. Anthocyanins are also molecules that have stability problems and environmental conditions, such as changes in pH, high temperatures, and oxygen presence, can easily deteriorate them [11].

The most common preservation methods for ingredients rich in natural antioxidants include dehydration by drying processes (conventional, solar, microwave, and vacuum drying), freeze-drying, and spray-drying microencapsulation [12,13,14]; however, these processes offer certain disadvantages. For example, for some antioxidant molecules that are subjected to temperatures above 120 °C, their degradation is accelerated. Tonon et al. [15] reported a decrease of up to 21% in the anthocyanin content of açai juice (*Euterpe oleracea*) when spray-drying was used, with tapioca starch as a bulking agent. Another problem is related to the acquisition and operating costs of the equipment required to carry out these processes.

On the other hand, nanocellulose (NC) is a natural polymer derived from cellulose; it receives this name when the cellulose breaks down, and small strands or nanocrystals are obtained. This phenomenon occurs thanks to the breaking of inter- and intramolecular hydrogen bonds, when used in different physical or chemical methods, resulting in at least one of the dimensions being on the nanometric scale [16]. NC is a material that has quickly gained attention thanks to its surface and mechanical properties, low density, and biocompatibility. In addition, there are a large number of plant sources, many of which are sustainable and renewable [17].

The berries of *H. macrocarpa* are an important source of bioactive substances such as polyphenols and anthocyanins [18]. It has been reported that this fruit has an anthocyanin content of 17.18 mg/g [19]. Despite this, the market for this fruit is not yet very established, and its consumption is mainly via fresh or processed products in an artisanal way, such as wine-type drinks, jams, jellies, and desserts.

Because of these antecedents, the aim was to obtain an anthocyanin-rich extract from ripe berries of *H. macrocarpa* and to load these antioxidants in the NC obtained from agro-industrial byproducts. Additionally, the storage stability of the powder at different temperatures was determined as a function of the content of anthocyanins, color, and antioxidant capacity. 

## 2. Results and Discussion

### 2.1. Content of Bioactive Substances of Different Hyeronima macrocarpa Extracts

The extraction of compounds with bioactive potential from plant sources is of great interest because it is the main stage for the correct use of these substances. The main objective of solvent extraction processes is to solubilize important compounds from the plant cell, to concentrate and purify them. The extraction of anthocyanins is carried out with polar and acidified solvents due to the greater stability in these conditions, mainly with mixtures of methanol and hydrochloric, formic, or acetic acid [20]. In this work, it was decided not to use methanol due to biosafety problems [21]. Instead, four commonly reported methodologies were used to extract anthocyanins and polyphenolic compounds. Table 1 shows the content of the main antioxidant metabolites that were obtained using different methods of extracting berries from *H. macrocarpa*. In general, the different extraction methods showed significantly different behaviors (*p* < 0.05) on the content of antioxidant metabolites.

#### 2.1.1. Effect of Extraction on the Content of Total Anthocyanins

Berries or red fruits are recognized by consumers as a source of beneficial substances for health [22]. This fact is supported by the different types of molecules that are part of their composition, such as anthocyanins. The different extraction methods of *H. macrocarpa* show anthocyanin contents ranging from 724 to 1317 mg C3G eq./100 g FW. HCl/ethanol (1:99 *v*/*v*) presented the highest anthocyanin extraction yield, followed by HCl/water (1:99 *v*/*v*), while ATPE and water (70 °C) presented a slightly lower yield. The solid–liquid extraction of phytochemicals from plants, using solvents, implies a distribution equilibrium or partition coefficient, which is defined as the ratio between the amount of the compound that is extracted by the solvent and the amount of same compound present in plant material. The characteristics of the solvent and the molecules of interest significantly affect the extraction process. However, other processes such as solvent diffusion through the material and interaction with other solutes must also be considered [23]. For the anthocyanins present in the fruits, the extraction with HCl/ethanol (1:99 *v*/*v*) was more efficient compared to the other extraction systems. One explanation would be that this solvent has a greater capacity and velocity to solubilize anthocyanins, rapidly reaching equilibrium concentration, due to a lower interfacial tension of ethanol compared to extraction systems that use water [24]. The results indicate that the extraction HCl/ethanol (1:99 *v*/*v*) constitutes a suitable method for obtaining these bioactives; it is even 50% more efficient than ATPE, a method that is typically used for its selectivity [25,26]. Other commercially recognized berries such as elderberry (*Sambucus nigra*) and chokeberry (Aronia spp.) presented values of 1375 and 1480 mg C3G eq./100 g FW, respectively [27]. On the basis of our results, it is possible to consider that *H. macrocarpa* berries are an important source of anthocyanins.

#### 2.1.2. Extraction Effect on the Polyphenol and Flavonoid Content

The effect of the different extractions on the total polyphenol and total flavonoid content of *H. macrocarpa* showed significant differences (*p* < 0.05) in most of the treatments evaluated. The water (70 °C) method presented the highest extraction for both polyphenols and flavonoids (Table 1). It has been widely reported that the increase in the extraction temperature offers the necessary kinetic conditions for the solvent to diffuse more quickly in the cellular tissues of the fruit, especially in the vacuoles, and solubilize a greater amount of compounds [28,29,30]. HCl/ethanol (1:99 *v*/*v*) and ATPE did not show significant differences (*p* > 0.05), yielding interesting amounts of total polyphenols. Similar yields using methanol as solvent were reported by Pérez-Jiménez et al. [31] and Guerrero et al. [32] in *Vaccinium angustifolium* berries and Maqui berries (*Aristotelia chilensis*), respectively. The flavonoids showed similar behavior, with the extraction using water (70 °C) and ATPE being the best methods.

#### 2.1.3. Effect of Extraction on Reducing Capacity and Free-Radical Scavenging

The antioxidant activity of *H. macrocarpa* extracts was determined using the FRAP method and the scavenging capacity of the main reactive oxygen species (hydroxyl, superoxide, and peroxyl radicals). The results showed that all extraction methods presented the ability to trap free oxygen radicals and reducing capacity. The extraction with water (70 °C) was the method that presented the highest activity in each of the evaluated tests, followed by ATPE, HCl/ethanol (1:99 *v*/*v*), and HCl/water (1:99 *v*/*v*) (Figure 1). This behavior correlates with the content of total polyphenols and flavonoids (Table 1), indicating that these are the metabolites responsible for exerting the antioxidant activity. The reducing capacity of polyphenols and flavonoids, measured by FRAP, can be explained by the ability of these molecules to chelate metals such as iron and copper, preventing these metals from subsequently generating free radicals [33,34]. 

The scavenging capacity of hydroxyl and superoxide radicals for the different H. macrocarpa berries extraction methods was also determined, since they are the most critical radical oxygen species. The results are presented in Figure 2. In line with what was found for FRAP, the water (70 °C) extraction was most effective in removing ROS, followed by ATPE, HCl/ethanol (1:99 *v*/*v*). and HCl/water (1:99 *v*/*v*). The reaction mechanisms involved in the scavenging of free radicals by polyphenols indicate that these molecules can prevent the problems caused by free radicals by trapping them directly. A flavonoid, for example, reacts with an ROS due to the high reactivity of the hydroxyl groups; the molecule is oxidized, generating a more stable radical and inactivating the injurious radicals. A study by Amić et al. [35] on metabolic derivatives of caffeic acid and ferulic acid showed that the presence of catechol, guaiacil, and carboxyl groups favored high efficiency in trapping free radicals, due to double mechanisms of electron transfer (2H^+^/2e^−^) via sequential proton loss (SPLET). In the case of anthocyanins, which are also polyphenols, the process is very similar; however, in this case, the predominant mechanism of antiradical capacity is directly related to the ease of these molecules in transferring hydrogen atoms to neutralize radicals and convert to resonance-stabilized and less reactive phenoxyl radicals [36]. These results suggest the ROS-scavenging activity was not significantly influenced by the extraction methods (*p* > 0.05), while the reducing capacity was significantly (*p* < 0.05) influenced by the extraction method and directly correlated with the content of antioxidant metabolites, specifically total polyphenols and flavonoids.

In general, berries have shown a remarkable antiradical capacity using in vitro and in vivo models. Cranberry (*Vaccinium macrocarpon*) and blueberry (*Vaccinium myrtillus*) showed high radical-scavenging properties in a xanthine/xanthine oxidase system that generates superoxide radicals, achieving IC_50_ values of 27 µg/mL for cranberry juice and 7 µg/mL for blueberry juice [37]. Assays on human hepatoma cells (HepG2) showed that acidified ethanol extracts from lingonberry (*Vaccinium vitis-idaea*) protected cells from oxidative damage caused by ROS generated by H_2_O_2_, confirming that the antioxidants present in berry-type fruits can protect cells from oxidative stress by acting synergistically [38].

### 2.2. Nanocellulose from Banana Pseudo-Stems

#### 2.2.1. Functional Group Analysis by FT-IR

The conversion of raw material to NC was followed by analyzing the functional groups using FT-IR. Figure 3 shows the infrared spectrum of untreated banana pseudo-stems, bleached cellulose, and NC obtained after chemical treatments. For the untreated material, characteristic absorption bands can be observed between ~1725 cm^−1^ and ~1735 cm^−1^, corresponding to stretching of the carbonyl group (C=O) that can be attributed to nonconjugated ketones [39] or the acetyl and uronic ester groups present in the hemicellulose molecule [40]; this band may also correspond to the ester group of the carboxylic group of phenolic acids (ferulic and p-coumaric) of the lignin molecule [41]. At ~1510 cm^−1^, a band can be observed in the spectrum of the raw material, corresponding to the vibrations of the stretching of the C=C bonds that are part of the benzene rings of the lignin molecule [42]. The thin band with average absorption at ~1629 cm^−1^ corresponds to the characteristic bending of the O–H, both of the different lignocellulosic polymers and of the adsorbed water molecules [43]. This band correlates with the strong absorption occurring in a broad band between ~3200 cm^−1^ and ~3600 cm^−1^, corresponding to the stretching vibration of the O–H bond [43]. The interaction with other OH groups in the cellulose molecules is reflected by a band at ~2920 cm^−1^ related to the C–H bond stretching vibrations [44]. Some characteristic absorption bands are present in the fingerprint region corresponding to 1031 cm^−1^, associated with the stretching vibration of the C–O–C group present in the heterocyclic ring of pyranose [44]. These bands are related to those of 1151 cm^−1^ and 1108 cm^−1^ ascribed to C–C stretching and glycosidic ether bonds [45]. The absence of bands in the regions of ~1725 cm^−1^ to ~1735 cm^−1^ and ~1510 cm^−1^ in the bleached cellulose and NC samples indicate the elimination of hemicellulose and lignin in the acid hydrolysis process. The three samples shared the characteristic adsorption bands of cellulose, such as the broad band at ~3300 cm^−1^ (stretching of the intramolecular hydrogen bonds of the O–H groups), small bands at ~1386 cm^−1^ indicating in-plane scissoring movements of CH_2_, and the C–H bending observed at ~1428 cm^−1^ [46]. The β-glycosidic bond between the glucose molecules that form cellulose can be observed at ~896 cm^−1^ [47]. 

#### 2.2.2. X-Ray Diffraction (XRD)

NC crystallinity can be defined as the ratio of crystalline NC to the total amount of material. In this work, the measurement was performed using X-ray diffraction, a technique that measures the phenomenon of photon dispersion that collides inelastically with atoms arranged periodically in crystal lattices and generates dispersed light rays with defined phase relationships. The X-ray diffraction patterns for NC are presented in Figure 4. In the diffractogram, a shoulder-shaped peak between 2θ = 12.5° and 18° is observed, corresponding to cellulose I (amorphous region) followed by a most prominent peak at 2θ = 18° to 27° corresponding to the crystalline region of cellulose II. 

The crystallinity index was calculated using Segal’s empirical method, where the heights of the peaks for the crystalline and amorphous cellulose zones are related. The NC from banana pseudo-stems showed a high crystallinity value, with a CI of 84.5%. The results suggest that the acid hydrolysis was very efficient, removing most amorphous structures such as lignins and hemicellulose. These data are slightly lower than those reported by Theivasanthi et al. [48], who obtained NC from cotton fibers with a crystallinity of 91.2%, using the same preparation method.

#### 2.2.3. Particle Sizes Using DLS

The NC obtained from banana pseudo-stems presented a unimodal particle size distribution, with sizes ranging from 71.0 nm to 212.5 nm, as observed in Figure 5. Most of the particles (90th percentile) presented sizes between 87.3 nm and 108.7 nm. With these results (added to those of FT-IR and XRD), it can be suggested that the production of NC from agro-industrial residues via acid hydrolysis was effective.

#### 2.2.4. Nanocellulose Surface Properties 

NC is a material that is characterized by having a large, very active surface porous area. The characterization of the porosity was carried out using two inert gases, N_2_ and CO_2_. With N_2_, the total BET (Brunauer–Emmett–Teller) area was measured, in addition to the size and distribution of large pores up to micropores; with CO_2_, the ultramicroporous area was measured. [49]. Table 2 summarizes the surface properties of NC obtained from banana pseudo-stems. The surface area measured using the BET method was 48.3 m^2^/g, and the ultramicroporous area was 6.2 m^2^/g, while the total porous volume was 0.28 cm^3^/g. Regarding the pore size distribution, a unimodal distribution was found with an average diameter of 8.2 nm. IUPAC classifies materials according to the size of the predominant pores [50]. The NC obtained in this work can be classified as a material that mainly presents narrow mesopores. The presence of this type of pore in NC is vital for applications as adsorbent materials, separation media, catalysts, and loading bioactive substances [51,52].

#### 2.2.5. TEM Microscopy

The morphology of banana pseudo-stem NC was analyzed using TEM, a technique that uses the dispersion of electrons deflected by the material to produce a grayscale image that allows observing the precise morphology of this material, with higher magnification and definition than other types of microscopes. The images obtained are presented in Figure 6. Broadly, different superimposed and chain-shaped structures could be observed, forming a kind of cluster. Performing a more detailed analysis of each micrograph, irregular to almost spherical structures were observed, which presented slightly uniform sizes between 87 nm and 124 nm. Generally, NC under the TEM microscope presents elongated shapes called rods or whiskers; however, in this work, a spherical or globose shape was mainly obtained, which indicates that the NC fibrils aggregated, forming a kind of sphere, or that the NC was detached from the material, due to the hydrolysis process, in the form of small nanometric-sized particles called nanocrystals. Similar results have been reported in agricultural residues of rice straw to produce drug-loaded capsules [53], while El-Hadi (2017) [54] also reported cellulose nanocrystals that had the same shape, used as reinforcement for food packaging. In line with the results obtained using TEM, XRD, DLS, and FT-IR, it can be concluded that the lignocellulosic material obtained via acid hydrolysis of banana pseudo-stems corresponded to NC, as indicated by the size, the crystallinity, and the functional groups.

### 2.3. Preparation of Hyeronima macrocarpa and Nanocellulose Powder

For the powder preparation, HCl/ethanol (1:99 *v*/*v*) extract was selected since it presented the highest content of anthocyanins compared to the other extraction methods (Table 1). In this research, simple impregnation was used as a tool to support natural antioxidants on NC. This is an easy method that does not require complex equipment and works very well for porous and highly adsorbent materials, such as NC extracted from banana pseudo-stems [55]. After the drying process, a powder with an intense and uniform magenta color was obtained (Figure 7). This indicates that, at first glance, the *Hyeronima macrocarpa* extract (purple) managed to be adsorbed by NC without appreciable deterioration of the anthocyanins, which are the main substances responsible for the color. According to the CIELAB color coordinates, the powder presented L* = 59.16, a* = 35.61, and b* = 7.08. These values are similar to those reported by Yamashita et al. [56] in lyophilized extracts rich in anthocyanins from blackberry byproducts loaded on maltodextrin.

#### 2.3.1. Thermogravimetric Analysis

The thermal decomposition of the powder was analyzed using thermogravimetry. The technique is based on monitoring the weight loss of a substance due to the formation of decomposition products that volatilize depending on the temperature of the medium. The thermogram is presented in Figure 8A. It can be observed that the powder did not present thermal decomposition up to 250 °C, and it is from this temperature onward that the degradation process occurred, reaching its maximum point at 340 °C. On the other hand, Figure 8B presents the thermogravimetric analysis of the *H. macrocarpa* extract that was used to load onto the NC. Above 100 °C, there was a considerable loss of mass potentially associated with water molecules and other low-molecular-weight volatiles in the extract.

Once 200 °C was reached, the most significant mass loss due to decomposition occurred, followed by another at 350 °C. The difference in the thermograms allows us to conclude that the extract loaded on the NC presented greater stability to thermal degradation, possibly due to the interactions between the polar molecules of the extract and the OH groups of the NC through hydrogen bonds.

#### 2.3.2. Powder Microstructure

The morphology of *Hyeronima macrocarpa* and NC powder was monitored using SEM microscopy. Figure 9 shows the microstructure of the powder formed; most of the particles had a spherical shape, and there were no agglomerations or cracks. The images obtained are similar to those reported by Yu and Lv [57] in extracts of rose (Rosa rugosa) microencapsulated in gum arabic and maltodextrin. It should be noted that the samples obtained by our team used a more straightforward technique of simple impregnation with subsequent conventional drying. Hence, it becomes an interesting alternative to spray-drying with consequent economic savings in processes and equipment. The size of the particles obtained ranged from 15 μm to 21 μm, with a uniform distribution. Because of this size, it can be explained that, at a macroscopic level, the powder did not present agglomerations and had an adequate flow, which is vital for powdered food products [58].

### 2.4. Antioxidants Released from the Powder

Once the powder formed by the incorporation of *H. macrocarpa* on NC was obtained, the anthocyanin and polyphenol release kinetics were evaluated. In Figure 10, the desorption process can be observed over time. This figure shows that about 85% of polyphenols and 68% anthocyanins were delivered to the medium during the first 60 min. The desorption equilibrium was reached between 100 and 120 min; after this time, up to 86% of the polyphenols and 91% of the anthocyanins desorbed from the polymeric matrix. This means that the process of incorporating natural extracts rich in antioxidants on NC is a reversible process, since these bioactive substances were not retained in the polymeric matrix. This process is known as physisorption and was reported in previously published studies by our team [55].

The anthocyanins presented a higher desorption speed compared to the polyphenols, a phenomenon that can be explained by their greater solubility in water, which was the selected desorption medium. After the desorption process, the NC could be used in a new antioxidant impregnation process; accordingly, the material did not suffer alterations when mixed with water, thus being able to comply with several regenerative cycles. The same property was observed in hydrogels prepared with NC as a functional ingredient; these aerogels had the ability to adsorb and desorb methylene blue, maintaining the adsorption capacity greater than 55% of the initial, after three cycles of adsorption–desorption [59]. These results indicate the versatility of uses and properties possessed by NC when used as a polymeric support material for bioactive substances.

### 2.5. Powder Stability

#### 2.5.1. Stability of Anthocyanins from *H. macrocarpa* and Nanocellulose Powder

The thermal stability of the powder formed by the incorporation of the H. macrocarpa extract on NC was performed by measuring the degradation of total anthocyanins, the change in reducing capacity, and color. The temperatures evaluated were 35 °C, 45 °C, and 55 °C, following an accelerated model that covers the possible storage temperatures of a powder product that does not require refrigeration. The degradation kinetics of anthocyanins are presented in Figure 11. The fit that best described the behavior of the data was a first-order reaction kinetic model (R^2^ values from 0.9413 to 0.9662, Table 3). Various authors have reported that the thermal deterioration of anthocyanins follows this type of behavior, regardless of the type of matrix, with degradation being faster at a higher temperature [60,61]. The degradation was dependent on temperature, being greater for 55 °C than for 45 °C and 35 °C. The kinetic constants k presented in Table 3 numerically verify this behavior over time.

The dependence of the rate of a chemical reaction on temperature is synthesized using the Arrhenius model or equation. This model was used to find the kinetic parameters of Ea, K, and t_1/2_ (Table 3), enabling us to understand the degree of effect of the temperature on the anthocyanins loaded on the NC and to estimate the degradation at different temperatures. The Ea or activation energy indicates the minimum amount of energy required by a system to start a process or to reach the active state of a reaction. Higher values of Ea are associated with a greater dependence on temperature during the degradation process of anthocyanins. In this study, an Ea value of 65.76 kJ/mol was obtained. Other reports showed higher activation energies. For example, a value of 135.83 kJ/mol was found in the blackberry juice degradation process [62], whereas, for blueberry juice, a value of 80.4 kJ/mol was presented [63]. In contrast, the half-life (t_1/2_) is a more helpful parameter since it is used at the industrial level to compare different products or processes for the preservation and storage of food matrices. This parameter indicates the time required to reach 50% of its initial value. Regarding anthocyanin stability, it indicates the time that elapses for the anthocyanin content to reach half of its initial value at a specific temperature. As expected, the half-life differed for each storage temperature, being higher at lower temperatures and presenting a lower value at the highest temperature of 55 °C (Table 3). Through the Arrhenius equation, it is possible to estimate the half-life at temperatures different from the study; for example, at 25 °C the powder would have a half-life of 727 days, which suggests that the loading process of the anthocyanin-rich extract of H. macrocarpa on NC isolated from banana pseudo-stems generated a considerable stability of these molecules. This fact can be explained by the interaction of the anthocyanins with the NC, through hydrogen bonds of the phenolic hydroxyls of the anthocyanins and the free OH groups of the NC [55].

#### 2.5.2. Color Stability

Products made from berries have vibrant colors that vary from blue to red tones, due to the anthocyanins that they naturally contain. Anthocyanins are natural colorants that also have a high antioxidant and antiradical capacity, which places them as bioactive substances. The main disadvantage is that they are very unstable and are easily degraded by different environmental conditions, e.g., the presence of oxygen, changes in pH and temperature [64]. *H. macrocarpa* and NC powder presented a very intense magenta initial color, with CIELAB coordinates of L* = 59.16 ± 0.11, a* = 35.61 ± 0.06, and b*= 7.08 ± 0.04. Consumer acceptance of this type of powdered product is initially determined by intense pink or red tones [65]. To know the color changes that the powdered product may undergo during storage, a stability evaluation was carried out at three different temperatures (35 °C, 45 °C, and 55 °C), and CIELAB coordinates were determined. From these data, the total color change or ∆E during storage was calculated. The results are summarized in Table 4, where it can be observed that there was a significant difference (*p* < 0.05) in the color variation over time for the same storage temperature. High values of ∆E indicate appreciable changes to the naked eye in the color of the samples, while values of ∆E greater than 5.0 are considered detectable changes in color [66]; accordingly, from days 84, 48, and 36, the samples presented changes in color at temperatures of 35 °C, 45 °C, and 55 °C, respectively. The samples stored at 55 °C began to exhibit a brown color from day 72 (L* = 47.11 ± 0.09; a* = 21.83 ± 0.04; b* = 14.47 ± 0.05). According to Patras et al. [67], some easily oxidizable phenolic acids can form quinones that react easily with anthocyanins and form condensation products, which could explain the appearance of the brown color of the powder.

The samples stored at 35 °C and 45 °C did not show an appreciable brown color during the storage period. These results indicate that the loss of color from magenta to brown is related to temperature; hence, at storage conditions below 35 °C, it is possible to estimate that the color would be stable for longer, due to the preservation of anthocyanins, as described in the previous section.

#### 2.5.3. Oxidative Stability Measured by FRAP

The antioxidant capacity of the powder stored at different temperatures was measured using FRAP, a very simple and sensitive methodology that is based on the ability of an antioxidant to react with the Fe^3+^–TPTZ complex and, thus, reduce iron to Fe^2+^, which forms a bright-blue complex that can be measured photometrically at 593 nm. The FRAP results are summarized in Figure 12. The behavior of the reducing capacity was similar in the first 120 days of storage for the three temperatures evaluated, with no statistically significant changes observed. Thereafter, a significant decline in the reducing capacity of the powder was observed. This situation has been frequently reported in thermal degradation tests of matrices containing anthocyanins [68]. Patras et al. [67] proposed that the thermal degradation of anthocyanins during storage can form different intermediates, including phenolic acids, which explains the slight increase in FRAP values. When the storage time exceeded 72 days, it was observed that the antioxidant capacity was reduced, indicating the degradation of phenolic compounds and other antioxidant or reducing substances.

## 3. Materials and Methods

### 3.1. Vegetal Material

*Hyeronima macrocarpa* berries were obtained in El Encano, city of Pasto (Nariño, Colombia), with a mean temperature of 11 °C and a height above the mean sea level of 2800 m. Ripe dark violet fruits were selected. The samples were transported to the laboratory in properly refrigerated airtight containers, washed with distilled water, and stored frozen at −20 °C until the different analyses were carried out.

### 3.2. Extract Preparation

The fruits were cut into halves and pulped manually. A quantity of pulp was adequately weighed and put into the different extraction systems. These methods are commonly used to obtain anthocyanin-rich and polyphenol-rich extracts from vegetable raw materials. Each of the extraction methods was carried out in triplicate in independent experiments.

#### 3.2.1. Extraction with HCl/Ethanol (1:99 *v*/*v*)

A 5 g portion of pulp was homogenized for 15 s in ultraturrax (IKA-Werk, Staufen, Germany) with 200 mL of acidified ethanol prepared according to the following proportion: HCl/ethanol (1:99 *v*/*v*). The mixture was left to stand for 12 h, after which it was filtered and concentrated in a vacuum rotary evaporator (Laborota 4011, Heidolph, Schwabach, Germany) under the following conditions: water bath at 45 °C, steam temperature from 40 to 45 °C, cooling bath at 18 °C, and rotation speed of 120 rpm, to a final volume of 100 mL [68]. This extraction was named ethanol 1%.

#### 3.2.2. Extraction with HCl/Water (1:99 *v*/*v*)

A 5 g portion of pulp was homogenized for 15 s in ultraturrax (IKA-Werk) with 200 mL of acidified water prepared according to the following proportion: HCl/water (1:99 *v*/*v*). The mixture was left to stand for 12 h, filtered, and concentrated in a vacuum rotary evaporator under the following conditions: water bath at 45 °C, steam temperature from 40 to 45 °C, cooling bath at 18 °C, and rotation speed of 120 rpm, to a final volume of 100 mL [69]. This extraction was named water 1%.

#### 3.2.3. Water at 70 °C

A 5 g portion of pulp was homogenized for 15 s in ultraturrax (IKA-Werk) with 200 mL of distilled water. The mixture was heated in a water bath at 70 °C ± 1 °C for 20 min. The mixture was filtered and concentrated in a vacuum rotary evaporator under the following conditions: water bath at 45 °C, steam temperature from 40 to 45 °C, cooling bath at 18 °C, and rotation speed of 120 rpm, to a final volume of 100 mL [70]. This extraction was named water (70 °C).

#### 3.2.4. Aqueous Two-Phase Extraction (ATPE)

A 5 g portion of pulp was homogenized for 1 h with 200 mL of an aqueous solution containing 30% (*v*/*v*) ethanol and 19% (*w*/*v*) ammonium sulfate, after which the mixture was filtered and concentrated in a vacuum rotary evaporator under the following conditions: water bath at 45 °C, steam temperature from 40 to 45 °C, cooling bath at 18 °C, and rotation speed of 120 rpm, to a final volume of 100 mL [71]. This extraction was named ATPE. 

### 3.3. Analysis of Bioactive Compounds

#### 3.3.1. Total Anthocyanin Content

The estimation of the total anthocyanin content was carried out using the pH differential method. This analysis consists of placing the different extracts in contact with two buffer solutions of pH 1.0 (a potassium chloride 25 mM solution adjust pH to 1.0 with HCl (37%, *v*/*v*)) and 4.5 (a sodium acetate 0.4 M solution adjust pH to 1.0 with (HCl 37%, *v*/*v*)). The change in absorbance was recorded at two wavelengths (530 nm and 700 nm) in a Multiskan Spectrum spectrophotometer (Thermo-Scientific, Waltham, MA, USA). The total anthocyanin content was calculated using Equation (1), and cyanidin-3-glucoside was used as a reference. The amount of total anthocyanins in the extracts was expressed as milligrams equivalent of cyanidin-3-glucoside/100 g of pulp [72].
(1)$$\frac{mg\;C3G}{100\;g}=\frac{A\times MW\times 1000}{\varepsilon \times L\times C},\;$$
where A = [pH 1.0 (A_530_ − A_700_) − pH 4.5 (A_530_ − A_700_)], MW = 449.2 g/mol (molecular weight of cyanidin-3-glucoside), ε = 26,900 L/(mol·cm) (molar extinction coefficient of cyanidin-3-glucoside), L = 1 cm (path of the light beam), and C is the concentration of the different extracts in g/L.

#### 3.3.2. Total Polyphenol Content (TPC)

The determination of the total polyphenolic content of *H. macrocarpa* extracts was carried out using the Folin–Ciocâlteau colorimetric method described by Singleton and Rossi (1965) [73]. In this assay, 50 µL of the samples were mixed with 125 µL of Folin–Ciocâlteau reagent and 425 µL of sodium carbonate solution at a concentration of 7.1% *w*/*v*. The final volume was adjusted to 1.0 mL with distilled water. The blank was prepared by substituting distilled water for Folin’s reagent. The reaction was kept in the dark for 60 min. Subsequently, the intensity of absorbance at 760 nm was determined using a Multiskan Spectrum spectrophotometer (Thermo-Scientific, Waltham, MA, USA). To determine the amount of polyphenols in the sample, a calibration curve was generated using gallic acid as standard (Sigma-Aldrich, purity >98.0%); the results were expressed in mg of gallic acid equivalent per 100 g of pulp (mg GAE/100 g FW).

#### 3.3.3. Total Flavonoid Content

Total flavonoid quantification was performed using the method proposed by Papoti et al. [74]. The method is based on forming a complex between flavonoids and aluminum chloride. A volume of 100 μL of the extracts was mixed with 30 μL of 5% (*w*/*v*) NaNO_2_, 30 μL of 10% (*w*/*v*) AlCl_3_, 200 μL of 1 M NaOH, and 640 μL of distilled water. Absorbance was measured at 510 nm on a Multiskan Spectrum (Thermo-Scientific, Waltham, MA, USA). The calibration curve was performed using different catechin dilutions as standard, and the results were expressed as mg of catechin equivalent per 100 g of pulp (mg catechin eq./100 g FW).

### 3.4. Free-Radical Scavenging and Reducing Capacity

#### 3.4.1. ORAC Assay (Oxygen Radical Absorbance Capacity)

The capacity to trap peroxyl radicals (ROO^●^) was performed using the fluorometric method reported by Prior et al. [75], under controlled conditions of temperature and pH (37 °C and 7.4, respectively). Fluorescence decay readings were performed at an excitation λ of 493 nm with an excitation slit of 10 nm, an emission λ of 515 nm, an emission slit of 15 nm, and a 1% attenuator. For the development of the technique, a volume of 30 μL of samples was added to 2920 µL of fluorescein (70 nM) in a phosphate buffer (75 mM, pH 7.4) and kept at 37 ± 1 °C in the holder of a LS55 Fluorescence Spectrometer (Perkin Elmer, Waltham, Massachusetts, USA). Then, 50 μL of AAPH (0.6 mol/L) was added, and the consumption of fluorescein, followed by fluorescence and the protective effect of the samples, was calculated using the differences in areas under the curve (AUC) of the fluorescein intensity decay between the control and the sample. Three readings were made for each sample, including a negative control (using a pH 7.4 buffer solution instead of sample). This was compared against the standard curve of the antioxidant Trolox. The result was expressed as μmol Trolox/100 g sample, according to Equation (2).
(2)$$ORAC=\frac{AUC-AUC^{{}^{\circ}}}{AUC_{Trolox}-AUC^{{}^{\circ}}}f\left[\frac{µM}{g/L}\right],\;$$
where *AUC* is the area under the curve for the sample, *AUC°* is the area under the curve for the control, *AUC_Trolox_* is the area under the curve for Trolox, and *f* is the ratio of the concentrations of Trolox and extracts.

#### 3.4.2. Superoxide Radical (O_2_^−●^)

Superoxide radicals are generated by the NADH/PMS system. The reaction mixture consisted of 15 µL of the different diluted extracts, 60 µL of NBT solution (156 µM), 60 µL of NADH solution (468 µM), and 165 µL of phenazine methosulfate (PMS) (10 µM). The reaction mixture was incubated at 37 °C for 5 min. After this time, the absorbance was measured at a wavelength of 560 nm. The ability to trap superoxide radicals was calculated according to Equation (3) [76].
(3)$$\%\;Superoxide\;radical\;activity=\left(\;\frac{A_{control}-A_{sample}\;}{A_{control}}\right)\times 100,$$
where *A_control_* is the reaction mixture replacing the sample with distilled water, and *A_sample_* corresponds to the reaction mixture with the sample.

#### 3.4.3. Hydroxyl Radical (OH^●^)

In vitro generation of the hydroxyl radical was carried out using the Fenton reaction involving Fe^2+^-EDTA/H_2_O_2_, during which peroxide decomposition to OH^●^ occurs [77]. These radicals react with terephthalic acid to form 2-hydroxyterephthalate acid, which has a high fluorescence. The reaction mixture consisted of 100 μL of the different extracts diluted in water, 300 μL of a 1 × 10^−^^4^ M sodium terephthalate solution, 2420 μL of phosphate buffer (0.2 M and pH 7.4), 90 μL of an EDTA (ethylenediaminetetraacetic acid disodium salt dihydrate) solution (1 × 10^−^^2^ M), and 90 μL of Fe^2+^ (1 × 10^−^^2^ M). The kinetics of the reaction was followed for 6 min in a LS-55 fluorometer (Perkin-Elmer, Waltham, MA, USA) at an excitation λ of 326 nm and an emission λ of 432 nm, with slits of 10 nm, without attenuation [78]. The results were presented as the ability to scavenge hydroxyl radicals, using Equation (4).
(4)$$\%\;Hydroxyl\;radical\;activity=\left(\;\frac{A_{2}-A_{0}\;}{A_{1}-A_{0}\;}\right)\times 100,$$
where *A*_0_ is the absorbance of the blank, *A*_1_ is the absorbance of the control (without sample), and *A*_2_ is the absorbance of the samples.

#### 3.4.4. Reducing Capacity by FRAP Method (Ferric Reducing Antioxidant Power) 

The reducing power of extracts was evaluated by measuring their ability to reduce ferric iron (Fe^+3^) complexed with TPTZ (2,4,6-tris(2-pyridyl)-s-triazine) to ferrous iron (Fe^+2^). For this, 50 µL of the different diluted extracts were taken and mixed with 900 µL of a solution of acetate buffer (0.1 M and pH 3.4), TPTZ solution (10 mM), and FeCl_3_ (20 mM), in a ratio of 10:1:1. After 30 min of reaction, the absorbance at a wavelength of 593 nm was measured. The results were expressed as TEAC equivalent (Trolox equivalent antioxidant capacity; µmol of Trolox/100 g FW), using Trolox as the reference antioxidant [79].

### 3.5. Nanocellulose from Banana Pseudo-Stem Waste

The pseudo-stems were washed, dried, and cut into pieces of 1 cm on each side. A total of 1 kg of the material was placed in a drying oven at 80 °C for 24 h. Once dry, the material was crushed and sieved to a size of 2 mm. A portion of 200 g of powdered material was treated with a 1.5 L of a mixture composed of hexane, ethanol, and ethyl acetate in a 2:1:1 ratio to eliminate the lipid components of the material, followed by treatment with 500 mL of a NaOH solution (10 %, *m*/*v*) for 7 h at 45°C. The biomass was washed with plenty of water and dried at 80 °C for a further 12 h. Then, the pulp was treated with a mixture of acetic and nitric acid (80% *v*/*v* and 65% *v*/*v*, respectively), in a ratio of 10:1 at 110 °C for 20 min with constant stirring in a hotplate stirrer (Benchmark Scientific, Sayreville, NJ, USA). The cellulose formed was separated from the acid solution by decantation and washed with a mixture of ethanol and water. Finally, the material was kept at 80 °C for 12 h of drying [80].

To transform cellulose into NC, the protocol proposed by Fahma et al. [81] was used. For this, 100 g of cellulose obtained was treated with 800 mL of sulfuric acid (64% *v*/*v*) for 60 min at 45 °C with constant stirring. Successive washes with distilled water were performed to remove excess acid, in addition to centrifugation for 20 min at 4 °C and 8000× *g*, and removal of supernatant acid water. When the pH reached 7.0, the suspension was placed in an ultrasound bath (GT SONIC) to facilitate its dispersion. After removing the remaining water, the NC was dehydrated at 105 °C for 12 h. The NC was characterized using FT-IR, DLS SEM, TEM, TGA, and XRD, surface area and volume, and pore size distribution analyses.

#### 3.5.1. FT-IR Spectroscopy

NC was analyzed by Fourier-transform infrared spectroscopy, using Spectrum-Two equipment (Perkin-Elmer, Waltham, MA, USA), with an Attenuated Total Reflection (ATR) device. The spectra were analyzed using Spectrum 10 software (Perkin-Elmer, Waltham, MA, USA) and were obtained in the wavenumber range of 4000–650 cm^−1^, with a resolution of 4 cm^−1^ [82]. 

#### 3.5.2. Particle Size Distribution Using DLS

Particle size distribution was determined by dynamic light scattering (DLS) using laser diffractometry (Nanoplus Zeta/Nano, Particulate Systems, Norcross, GA, USA), under the following conditions: particle refractive index, 1.59; absorption coefficient of particles, 0.01; water index of refraction, 1.33; water viscosity, 0.8872 cP; temperature, 25 °C, with a general calculation model for irregular particles. Material suspensions from 5 to 25 mg/L were prepared using distilled water as the solvent, which was kept under ultrasound for 25 min. Measurements were performed in triplicate. The calculations and reporting of the results were performed in the NanoPlus software (Version 5.23/3.00, Micromeritics, Norcross, GA, USA).

#### 3.5.3. SEM Microscopy

The morphology of the NC was analyzed by scanning electron microscopy in a NanoSEM 430 equipment (FEI, Thermo-Fischer Scientific, Hillsboro, OR, USA) adjusted to an acceleration voltage of 15 kV. A drop of the NC suspensions (0.2% *w*/*v*) was placed on a glass grid and vacuum-dried prior to SEM analysis. Once the solvent was evaporated, it was plated with gold on a sputter coater. Different areas of the grid were evaluated at different magnifications. The best ones for imaging were chosen [83].

#### 3.5.4. TEM Microscopy

The morphology of the NC was analyzed using a transmission electron microscope (TEM; FEI, Tecnai G2 Thermo-Fischer Scientific, Hillsboro, OR, USA), operated at an energy of 140 keV. The samples were dispersed in ethanol and then exposed to ultrasound for 30 min, to facilitate the material disintegration. A drop of dispersion was placed on a nickel grid, and then uranyl acetate (2% *w*/*v*) was dropped onto the same grid. After 5 min of drying, the analysis was performed [84].

#### 3.5.5. Thermogravimetric Analysis (TGA)

The thermal stability of the NC samples, isolated from banana pseudo-stems, was analyzed using TGA/DSC equipment (Q50, TA Instruments, Inc., New Castle, DE, USA). Samples were dried overnight in a vacuum oven before analysis began. A portion of 5.0 mg of sample was analyzed under nitrogen atmosphere at a flow rate of 20 mL/min. The heating temperature was between 30 and 750 °C, at a rate of 10 °C/min. The TGA equipment was calibrated using nickel as the reference material [85].

#### 3.5.6. Surface Area and Pore Volume 

In a TriStar II PLUS gas sortometer, the surface area and pore volume were obtained. The first was determined using the BET method (Brunauer–Emmett–Teller), and the second was determined through the integration of the NLDFT (nonlocal density functional theory) pore size distribution. For the analyses, the samples were previously degassed at a temperature of −60 °C in the presence of N_2_ for 24 h [86].

#### 3.5.7. X-Ray Diffraction (XRD)

Wide-angle XRD data were collected using an XPert Panalytical Empyrean Series II-Alpha1 X-ray (Malvern Panalytical, Malvern, UK) diffractometer equipped with Cu/K radiation at 30 kV and 15 mA to investigate the XRD spectra of the NC sample. Scattered radiation was detected in the 2θ range of 5°–60°, with a step size of 0.05° and 50 s per step. The degree of crystallinity was determined according to the empirical peak height method developed by Segal (1959); the apparent crystallinity of the material (as a percentage) was calculated from the height ratio between the intensity of the crystalline peak and the total intensity, after the subtraction of the background signal (non-crystalline) according to Equation (5) [44].
(5)$$IC=100\times \frac{I_{200}-I_{nc}\;}{I_{200}},$$
where *IC* expresses the percentage of apparent crystallinity, *I*_200_ is the maximum intensity of the peak corresponding to the plane in the sample with Miller 200 indices at 2θ = 22°–24°, and *I_nc_* represents the diffraction intensity of the non-crystalline material, which is taken at 2θ ≈ 18° in the valley between the peaks.

### 3.6. Loading of H. macrocarpa Bioactives onto Nanocellulose Using the Incipient Impregnation Method

A weight of 25 g of NC was heated at 105 °C for 2 h in a forced convection oven to remove the water. When the NC reached room temperature, it was uniformly dispersed at the bottom of a 1 L beaker, and 100 mL of a solution of *H. macrocarpa* extract at 250 mg/mL was very slowly added dropwise. The mixture was continuously stirred to avoid the formation of lumps. After the dripping process, the mixture was left overnight in an oven at 60 °C to eliminate as much solvent as possible. The following day, the powder formed was scraped from the bottom of the beaker and stored in a container for subsequent analyses.

#### Color Assay 

CIELAB parameters (L* = lightness, a* = red/green value, b* = blue/yellow value) were determined for the *H. macrocarpa* and NC powder, using a sphere spectrophotometer SP-62 (X-Rite, Grand Rapids, MI, USA), equipped with D65 illuminant at an observer angle of 10°. The total color change (ΔE) was determined using reference values of L*, a*, and b* of the freshly prepared powder and compared with color of the powder samples storage at different times and temperatures. The ΔE values were calculated according to Equation (6).
(6)$$\Delta E=\sqrt{{\left(L^{*}-L_{0}^{*}\right)}^{2}+{\left(a^{*}-a_{0}^{*}\right)}^{2}+{\left(b^{*}-b_{0}^{*}\right)}^{2}}\;$$
where L^*^_0_, a^*^_0_, and b^*^_0_ are the color powder values of the initial conditions. 

### 3.7. Delivery of Bioactives

To know the release profile of the bioactive substances of *H. macrocarpa* extract supported on NC, 100 mg of the previously prepared powder (Section 3.6) was taken and mixed with 10.0 mL of distilled water, and aliquots of 100 μL were taken every 2 min until the concentration reached equilibrium. Release kinetics were carried out at 25 °C. The release tests were expressed as a percentage of anthocyanins and polyphenols, according to Equation (7).
(7)$$\%\;release\;compounds=\left[\frac{Cmax-Cd}{Cmax}\right]\times 100,\;$$
where *Cmax* is the concentration of polyphenols or anthocyanins in the powder in mg/L, and *Cd* is the concentration of released compounds in mg/L. The assays were repeated in triplicate in independent experiments.

### 3.8. Stability Assay of H. macrocarpa and Nanocellulose Powder

#### 3.8.1. Kinetic Studies of Degradation of *H. macrocarpa* and Nanocellulose Powder

The change in the physicochemical attributes of the powder can be measured by the appearance or disappearance of a quantifiable parameter *P*, i.e., the content of total anthocyanins, FRAP antioxidant power, and powder color in this study. The degree of appearance or disappearance of *P* is represented according to Equation (8).
(8)$$r_{p}=-\frac{dP}{dt}=K{\left[P\right]}^{m},$$
where *K* is the rate constant, and *m* is the apparent reaction order.

Depending on the value that *m* takes (0, 1, or 2), Equation (6) can be converted into Equations (9)–(11).
(9)$$P=P_{o}-Kt\;m=0.$$
(10)$$P=P_{o}e^{-Kt}\;m=1.$$
(11)$$\frac{1}{P}=\frac{1}{P_{o}}-Kt\;m=2.$$

To establish the reaction order, the *p* value for each parameter was plotted as a function of time, and the most useful mathematical model representing the degradation kinetics of H. macrocarpa powder was obtained [87].

#### 3.8.2. Effect of Temperature on the *H. macrocarpa* Nanocellulose Powder Degradation 

The powder stability was measured under controlled storage conditions at 35 °C ± 1 °C, 45 °C ± 1 °C, and 55 °C ± 1 °C. Anthocyanin quantification, FRAP, and CIELAB color parameters were used as stability parameters. At a specific time, a portion of the powder was taken and analyzed. The estimation of stability as a function of temperature was carried out using the Arrhenius equation (Equation (12)) and its linearized form (Equation (13)).
(12)$$K=K_{A}\;e^{\left(\frac{-E_{a}}{RT}\right)},$$
(13)$$lnK=lnK_{A}-\frac{E_{a}}{RT},$$
where *K* is the rate constant, *K_A_* represents the constant of the Arrhenius equation (days^−1^), *E_a_* is the activation energy (J/mol), *R* is the universal gas constant (8.3144 J/mol·K), and *T* is the absolute temperature (K). To estimate the effect of temperature on reaction rate, *K* values at different temperatures over the range of interest were calculated, whereby −*ln*(*K*) was plotted against 1/*T* in a semi-log plot. A straight line with slope *E_a_*/*R* was obtained [87].

The half-life, defined as the time necessary for the *P* parameter to decrease to 50% of its initial value, was calculated using Equation (14).
(14)$$t_{1/2}=\frac{\ln2}{K}.$$

### 3.9. Statistical Analysis

Each analysis was performed in triplicate, and the results are presented as the mean ± standard deviation. Statistically significant differences were determined by one-way analysis of variance (ANOVA) with Student’s t-test; *p*-values <0.05 were considered statistically significant (STATGRAPHICS Centurion XVII (V17.2.07) software, Statgraphics Technologies Inc., The Plains, VA, USA).

## 4. Conclusions

The extracts of *Hyeronima macrocarpa* berries presented a significant content of antioxidant metabolites. The extraction with water (70 °C) offered the best conditions for obtaining polyphenols and flavonoids, while the extraction with HCl/ethanol (1:99 *v*/*v*) was more efficient with anthocyanins. The antioxidant capacity measured using various techniques also showed better results with the water (70 °C) extraction, closely followed by HCl/ethanol (1:99 *v*/*v*).

Since anthocyanins are highly valued metabolites for their bioactive and coloring properties, a powder was prepared by incorporating HCl/ethanol (1:99 *v*/*v*) extract on nanocellulose that was successfully extracted from banana pseudo-stems. The powder product presented an intense magenta color that indicates adequate incorporation of the extract in the porous material, in addition to important protection of the bioactives from thermal deterioration, measured by thermogravimetry. The powder presented a capsular morphology indicating that the nanocellulose was capable of adsorbing the substances via simple impregnation and the formation of microcapsules similar to those obtained in a spray-drying process. The stability tests over time at different storage temperatures showed that, at 35 °C, the anthocyanins in the powder had a half-life of more than 300 days, while color stability remained unchanged for up to 4 months.

The extract rich in anthocyanins managed to increase the shelf-life during storage, due to the impregnation process on the nanocellulose, which, due to its surface properties, protected the bioactive substances from the degradation process. In future research, it is necessary to evaluate the potential of the powder as an ingredient or food additive. Additionally, in vitro and in vivo bioaccessibility tests are recommended to evaluate the bioavailability that this system of antioxidant incorporation on nanocellulose can offer. 

## Figures and Tables

**Figure 1 molecules-27-06661-f001:**
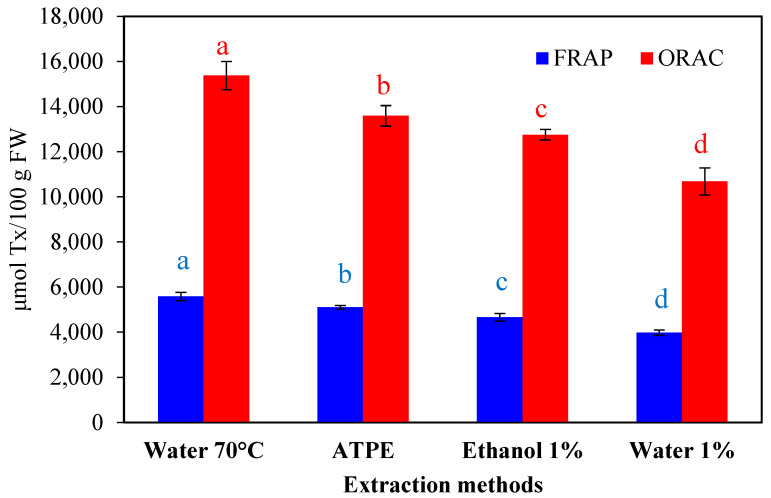
Antioxidant capacity of *H. macrocarpa* extracts measured in terms of FRAP and ORAC. Bars with different letters (a to d) at the same color, are significantly different (*p* < 0.05).

**Figure 2 molecules-27-06661-f002:**
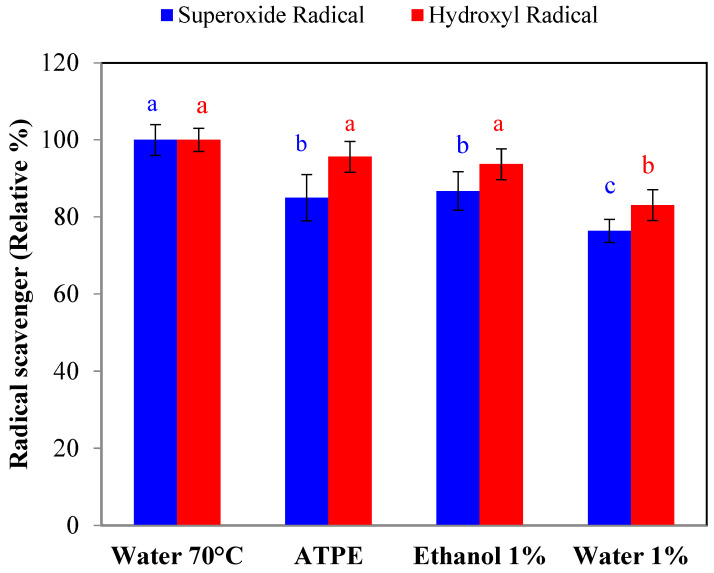
Radical-scavenging capacity of *H. macrocarpa* extracts. Bars with different letters (a to c) at the same color, are significantly different (*p* < 0.05).

**Figure 3 molecules-27-06661-f003:**
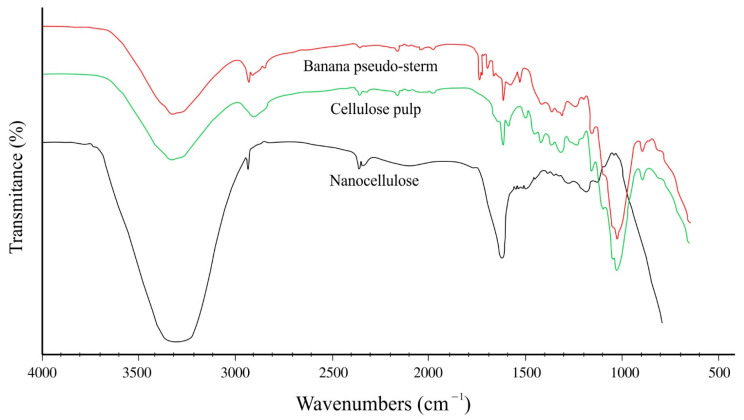
FT-IR spectra of nanocellulose, cellulose pulp, and banana pseudo-stem.

**Figure 4 molecules-27-06661-f004:**
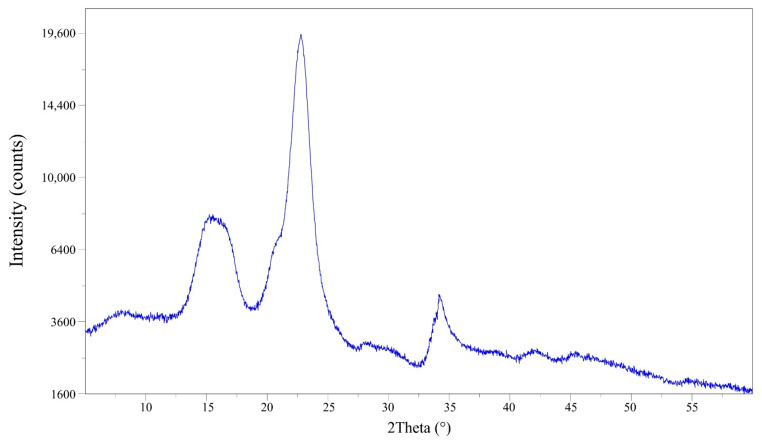
Diffractogram of nanocellulose isolated from banana pseudo-stems.

**Figure 5 molecules-27-06661-f005:**
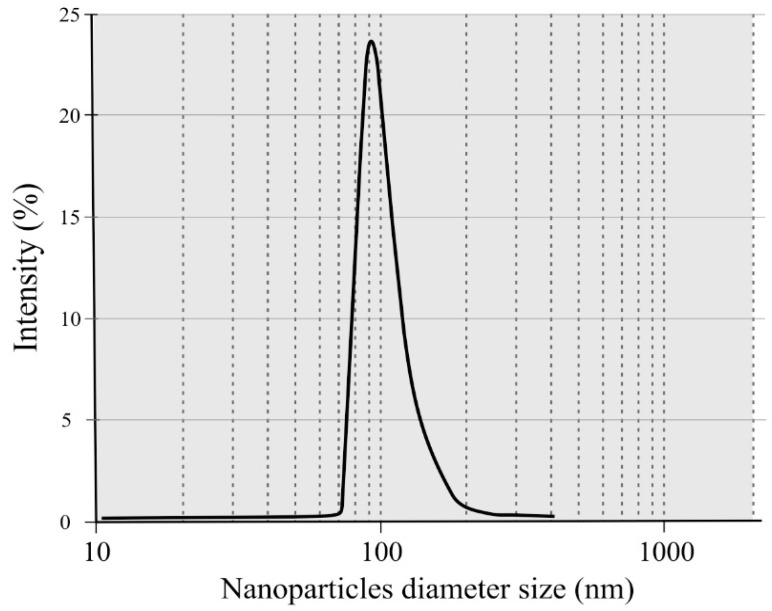
Particle size distribution according to DLS of nanocellulose isolated from banana pseudo-stems.

**Figure 6 molecules-27-06661-f006:**
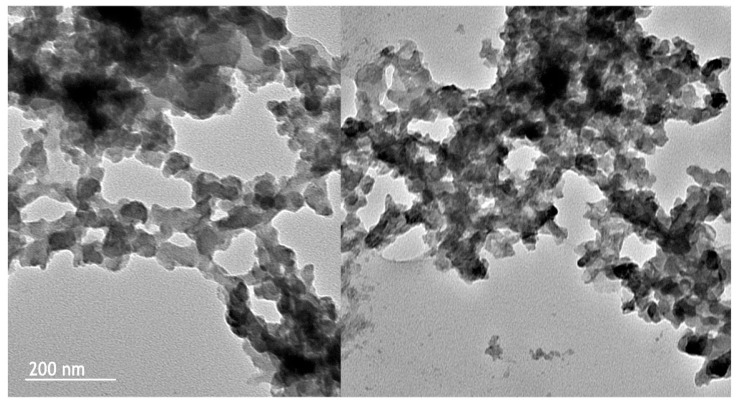
TEM microscope images of nanocellulose isolated from banana pseudo-stems.

**Figure 7 molecules-27-06661-f007:**
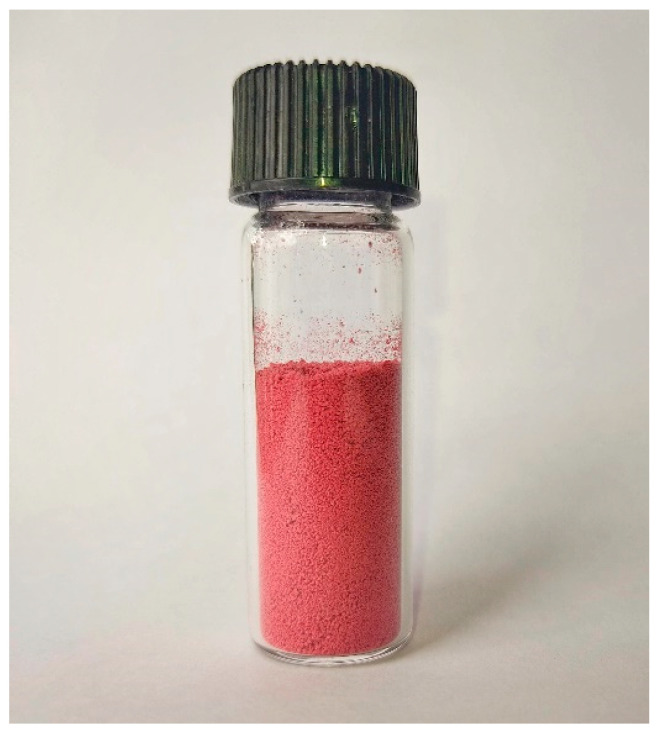
Photograph of *H. macrocarpa* and nanocellulose powder.

**Figure 8 molecules-27-06661-f008:**
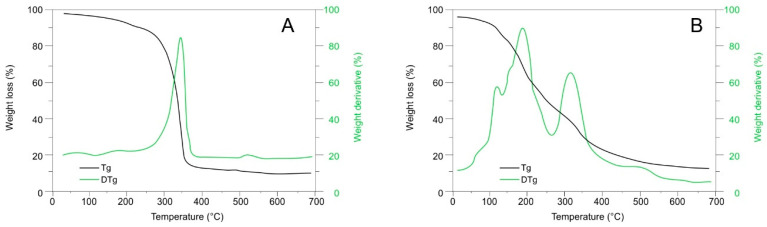
Thermogravimetric analysis of nanocellulose and *H. macrocarpa* powder (**A**) and *H. macrocarpa* extract (**B**).

**Figure 9 molecules-27-06661-f009:**
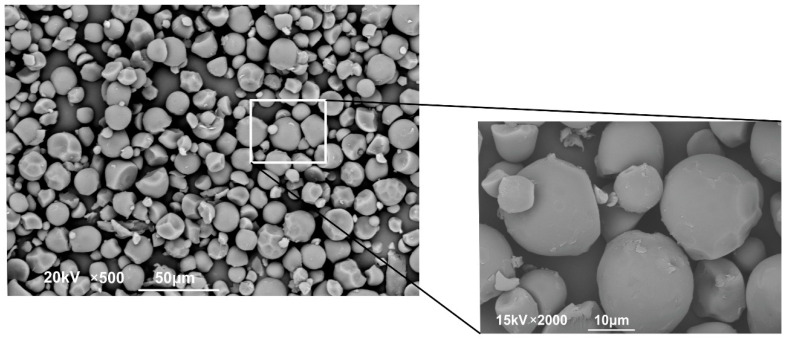
Micrographs of the powder formed following incorporation of the extract of *H. macrocarpa* on nanocellulose extracted from banana pseudo-stems: 2000× and 500× magnification.

**Figure 10 molecules-27-06661-f010:**
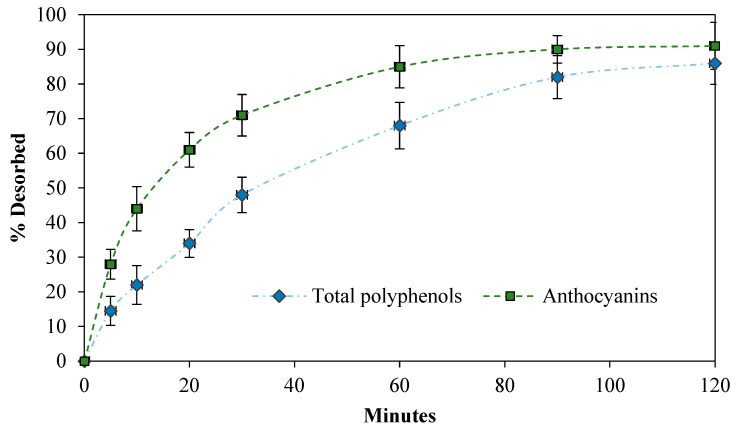
Release profile of polyphenols and anthocyanins from powder formed by the incorporation of *H. macrocarpa* on nanocellulose.

**Figure 11 molecules-27-06661-f011:**
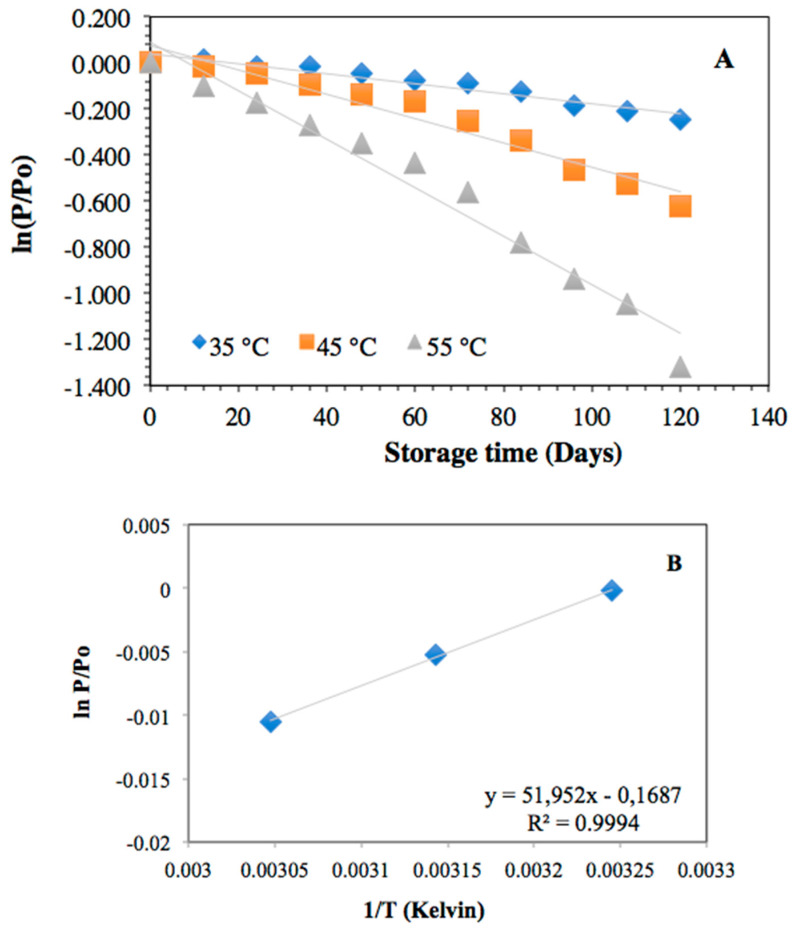
Thermal degradation kinetics at various temperatures of anthocyanins from *H. macrocarpa* powder and nanocellulose (**A**); Arrhenius plot of the degradation model during heating (**B**).

**Figure 12 molecules-27-06661-f012:**
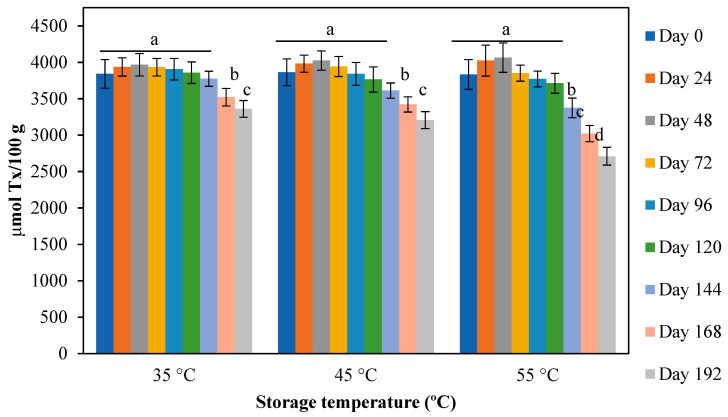
Effect of storage time at different temperatures on the antioxidant capacity measured using FRAP of *H. macrocarpa* and nanocellulose powder. Bars with different letters (a to d) for the each temperature treatment, are significantly different (*p* < 0.05).

**Table 1 molecules-27-06661-t001:** Content of antioxidant bioactive substances of *H. macrocarpa* berries.

Extraction Methods	Total Anthocyanins (mg C3G */100 g FW)	Total Polyphenols (mg GAE **/100 g FW)	Total Flavonoids (mg Catechin Eq./100 g FW)
Ethanol 1%	1317.39 ± 17.18 ^a^	859.47 ± 12.03 ^a^	173.76 ± 6.43 ^a^
Water 1%	969.42 ± 6.71 ^b^	810.01 ± 9.62 ^b^	133.29 ± 9.23 ^b^
ATPE	824.38 ± 12.74 ^c^	861.48 ± 18.02 ^a^	198.34 ± 11.64 ^c^
Water 70 °C	724.32 ± 9.56 ^d^	929.85 ± 12.97 ^c^	209.90 ± 4.78 ^c^

Data are the mean ± SD. Values in the same column followed by different letters (a to d) are significantly different (*p* < 0.05). * C3G: cyanidin-3-glucoside. ** GAE: gallic acid equivalent.

**Table 2 molecules-27-06661-t002:** Characterization of nanocellulose isolated from banana pseudo-stem.

Parameter	Value
BET specific surface area measured with N_2_	48.3 ± 5.1 m^2^/g
Ultramicroporous area measured with CO_2_	6.2 ± 0.4 m^2^/g
Particle diameter measured using DLS (90%)	87.3–108.7 nm
Particle size measured using TEM	87−124 nm
Mean pore diameter	3.8 ± 0.2 nm
Total pore volume	0.28 cm^3^/g

Data are presented as the mean ± SD.

**Table 3 molecules-27-06661-t003:** Kinetic parameters of anthocyanin and reducing capacity during heat treatment.

Parameter	Temp.	*k* × 10^3^ (days^−1^)	R^2^	*Ea* (kJ/mol^−1^)	t_1/2_ (days)
Total anthocyanins	35 °C	2.20 ± 0.13 ^a^	0.9413	65.76 ± 4.34 (R^2^: 0.9736)	315 ± 16 ^a^
	45 °C	5.30 ± 0.38 ^b^	0.9495		131 ± 7 ^b^
	55 °C	10.50 ± 0.97 ^c^	0.9662		66 ± 4 ^c^
FRAP	35 °C	0.40 ± 0.02 ^a^	0.9737	46.44 ± 2.71 (R^2^: 0.9707)	1733 ± 104 ^a^
	45 °C	1.00 ± 0.08 ^b^	0.9829		693 ± 32 ^b^
	55 °C	1.30 ± 0.07 ^c^	0.9061		533 ± 17 ^c^

Different letters in the same column indicate significant differences (*p* < 0.05).

**Table 4 molecules-27-06661-t004:** Effect of temperature on the total color difference (*∆E*) of *H. macrocarpa* and nanocellulose powder.

Storage Time (days)	35 °C	45 °C	55 °C
12	0.72 ± 0.01	1.00 ± 0.04	3.60 ± 0.02
24	1.73 ± 0.02	1.96 ± 0.01	6.62 ± 0.05
36	2.66 ± 0.02	3.69 ± 0.08	11.03 * ± 0.03
48	3.37 ± 0.04	5.54 * ± 0.05	14.23 * ± 0.07
60	4.28 ± 0.03	7.26 * ± 0.03	17.04 * ± 0.02
72	4.75 ± 0.05	8.19 * ± 0.06	19.40 ** ± 0.07
84	6.32 * ± 0.04	10.01 * ± 0.04	20.20 ** ± 0.11
96	9.14 * ± 0.06	11.05 * ± 0.08	21.36 ** ± 0.11
108	11.43 * ± 0.03	13.89 * ± 0.11	22.45 ** ± 0.13
120	11.31 * ± 0.11	14.73 * ± 0.09	23.38 ** ± 0.08

All storage time, for each temperature, presented significant differences (*p* < 0.05). *Appreciable color changes to the human eye. ** Presence of brown color.

## Data Availability

The data supporting the reported results can be found at the Laboratorio Ciencia de los Alimentos of the Universidad Nacional de Colombia and can be obtained by contacting the authors.

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
