# Peer review of "Antioxidants from Hyeronima macrocarpa Berries Loaded on Nanocellulose: Thermal and Antioxidant Stability"

_molecules, 2022, doi:10.3390/molecules27196661_

Round 1

Reviewer 1 Report

Review’s Report

Manuscript ID: molecules-1932387

Title: Antioxidants from Hyeronima macrocarpa Berries Loaded on Nanocellulose: Thermal and Antioxidant Stability

Journal: Molecules

In this manuscript, the authors have shown the effect of different storage temperatures ranging from 35°C to 55°C on bioactive substances and antioxidant properties of Hyeronima macrocarpa berries loaded on nanocellulose. It appears that nanocellulose, extracted from banana pseudo-stems, presents a rather interesting surface with high porosity properties. In this study, the content of bioactive substances of different Hyeronima macrocarpa extracts, the effect of extraction on the content of total anthocyanins, the effect on the polyphenols and flavonoids content, the effect of extraction on the reducing capacity and scavenging free radicals were successfully presented. Also, the applied instrumental techniques such as FT-IR, XRD, DLS, TEM, SEM, and TGA were carefully chosen.

Review’s suggestions

-          Line 383. Table 2, should be renamed Table 3.  Also, decimal commas should be replaced with decimal dots.

-          Line 415. Table 2, should be renamed Table 4.  Also, decimal commas should be replaced with decimal dots.

This study has the potential to be cited. 

I recommend to the Editorial Office to consider this manuscript for publication but after minor revision.

Author Response

Response to Reviewer 1 Comments

Journal: Molecules

In this manuscript, the authors have shown the effect of different storage temperatures ranging from 35°C to 55°C on bioactive substances and antioxidant properties of Hyeronima macrocarpa berries loaded on nanocellulose. It appears that nanocellulose, extracted from banana pseudo-stems, presents a rather interesting surface with high porosity properties. In this study, the content of bioactive substances of different Hyeronima macrocarpa extracts, the effect of extraction on the content of total anthocyanins, the effect on the polyphenols and flavonoids content, the effect of extraction on the reducing capacity and scavenging free radicals were successfully presented. Also, the applied instrumental techniques such as FT-IR, XRD, DLS, TEM, SEM, and TGA were carefully chosen.

Response,

We thank the reviewer for the time and effort in reviewing our work.

Review’s suggestions

Line 383. Table 2, should be renamed Table 3. Also, decimal commas should be replaced with decimal dots.

Line 415. Table 2, should be renamed Table 4. Also, decimal commas should be replaced with decimal dots.

Response,

We thank the reviewer for the correction, the respective changes in Tables 3 and Table 4 were made in the amended version of the manuscript.

This study has the potential to be cited.

I recommend to the Editorial Office to consider this manuscript for publication but after minor revision.

Response,

Many thanks to the reviewer for the trust given in our work, and again we appreciate your kind review.

Reviewer 2 Report

I have carefully read the article. I found this topic's contents, the experimental design, and the results exciting, but I have some comments that I have attached as "a word file" that you need to download and use to improve the manuscript. A lot of grammatical errors. I have corrected all the grammatical errors and highlighted the ones I have corrected as " red color" in the document I have attached. I am willing to review this article again after you incorporate my comments. Attached are my comments

Author Response

We especially thank the referee for the careful review of our work and the trust given.

We also appreciate the correction of grammatical errors and the contributions made to the manuscript. All of them were incorporated into the corrected version of the document.

We carefully review each of the comments and make the necessary modifications. In the attached file we present the most important ones.

Reviewer 3 Report

The manuscript entitled Antioxidants from Hyeronima macrocarpa Berries Loaded on Nanocellulose: Thermal and Antioxidant Stability” of ID: molecules-1932387 is relevant for publication in Molecules by MDPI after minor revision.

The subject of the article and the research results obtained have been presented and discussed in a clear and precise manner, not raising any doubts. The article refers to the promising health-promoting effects of Hyeronima macrocarpa berries, which have not been presented so far, and the reviewed article is of great cognitive value. Unfortunately, the methodological part of the article lacks information on the plant material used for the research, its confirmed authenticity and adequate voucher specimens.

Detailed comments:

Table 1 does not include the units of the test results it contains.

Author Response

Response to Reviewer 3 Comments

The manuscript entitled ”Antioxidants from Hyeronima macrocarpa Berries Loaded on Nanocellulose: Thermal and Antioxidant Stability” of ID: molecules-1932387 is relevant for publication in Molecules by MDPI after minor revision.

The subject of the article and the research results obtained have been presented and discussed in a clear and precise manner, not raising any doubts. The article refers to the promising health-promoting effects of Hyeronima macrocarpa berries, which have not been presented so far, and the reviewed article is of great cognitive value. Unfortunately, the methodological part of the article lacks information on the plant material used for the research, its confirmed authenticity and adequate voucher specimens.

Answer,

We are very grateful to the reviewer for his trust and kind comments. Regarding the plant material, we inform you that this plant is used in the wood production and use of the fruits, it is a plant commonly cultivated in the region where we carry out the collection process, and different taxonomists and government entities have already carried out their identification and classification, including the University of Nariño, which has jurisdiction over the area. The description of the species can be reviewed in the following document, which is in spanish: https://sired.udenar.edu.co/7341/

Detailed comments:

Table 1 does not include the units of the test results it contains.

Answer,

We appreciate your kind comment, the units for the content of antioxidant bioactive substances of H. macrocarpa berries, were included in Table 2 in the amended version of the manuscript.

Round 2

Reviewer 2 Report

The revised manuscript that has been submitted shows that a great deal of time and effort has been put into its preparation, and it is absolutely a better manuscript than the previous one. It is generally well-written, and I believe it deserves to be published in this journal. I have attached a file for a minor revision to improve the paper's readability, after which I recommend it be accepted for publication. Thanks

Author Response

The authors sincerely thank the reviewer for their kind suggestions and attention to detail in reviewing the manuscript. All of this helped us to improve this work.

New changes in the manuscript can be seen in red.
